# Melanogenesis Is Directly Affected by Metabolites of Melatonin in Human Melanoma Cells

**DOI:** 10.3390/ijms241914947

**Published:** 2023-10-06

**Authors:** Jack K. S. Möller, Kinga Linowiecka, Maciej Gagat, Anna A. Brożyna, Marek Foksiński, Agnieszka Wolnicka-Glubisz, Elżbieta Pyza, Russel J. Reiter, Meri K. Tulic, Andrzej T. Slominski, Kerstin Steinbrink, Konrad Kleszczyński

**Affiliations:** 1Department of Dermatology, University of Münster, Von-Esmarch-Str. 58, 48149 Münster, Germany; j_moel40@uni-muenster.de (J.K.S.M.); kerstin.steinbrink@ukmuenster.de (K.S.); 2Department of Human Biology, Faculty of Biological and Veterinary Sciences, Nicolaus Copernicus University, Lwowska 1, 87-100 Toruń, Poland; klinowiecka@umk.pl (K.L.); anna.brozyna@umk.pl (A.A.B.); 3Phillip Frost Department of Dermatology & Cutaneous Surgery, University of Miami Miller School of Medicine, Miami, FL 33125, USA; 4Department of Histology and Embryology, Collegium Medicum in Bydgoszcz, Nicolaus Copernicus University in Toruń, 85-092 Bydgoszcz, Poland; mgagat@cm.umk.pl; 5Department of Clinical Biochemistry, Faculty of Pharmacy, Collegium Medicum in Bydgoszcz, Nicolaus Copernicus University in Toruń, 85-092 Bydgoszcz, Poland; marekf@cm.umk.pl; 6Department of Biophysics and Cancer Biology, Faculty of Biochemistry, Biophysics and Biotechnology, Jagiellonian University, Gronostajowa 7, 30-387 Krakow, Poland; a.wolnicka-glubisz@uj.edu.pl; 7Department of Cell Biology and Imaging, Institute of Zoology and Biomedical Research, Jagiellonian University, Gronostajowa 9, 30-387 Kraków, Poland; elzbieta.pyza@uj.edu.pl; 8Department of Cell Systems and Anatomy, UT Health, Long School of Medicine, San Antonio, TX 78229, USA; reiter@uthscsa.edu; 9Team 12, INSERM U1065, Centre Méditerranéen de Médecine Moléculaire (C3M), Université Côte d’Azur, 06200 Nice, France; meri.tulic@unice.fr; 10Department of Dermatology, Comprehensive Cancer Center, University of Alabama at Birmingham, Birmingham, AL 35294, USA; aslominski@uabmc.edu; 11Pathology and Laboratory Medicine Service, VA Medical Center, Birmingham, AL 35294, USA

**Keywords:** melatonin, melanogenesis, kynurenic and indolic metabolites, tyrosinase, human melanoma, G-protein-coupled membrane receptors, luzindole, molecular mechanism

## Abstract

Melatonin (*N*-acetyl-5-methoxytryptamine, MEL), its kynurenic (*N*^1^-acetyl-*N*^2^-formyl-5-methoxykynurenine, AFMK) and indolic derivatives (6-hydroxymelatonin, 6(OH)MEL and 5-methoxytryptamine, 5-MT) are endogenously produced in human epidermis. Melatonin, produced by the pineal gland, brain and peripheral organs, displays a diversity of physiological functions including anti-inflammatory, immunomodulatory, and anti-tumor capacities. Herein, we assessed their regulatory effect on melanogenesis using amelanotic (A375, Sk-Mel-28) and highly pigmented (MNT-1, melanotic) human melanoma cell lines. We discovered that subjected compounds decrease the downstream pathway of melanin synthesis by causing a significant drop of cyclic adenosine monophosphate (cAMP) level, the microphthalmia-associated transcription factor (MITF) and resultant collapse of tyrosinase (TYR) activity, and melanin content comparatively to *N*-phenylthiourea (PTU, a positive control). We observed a reduction in pigment in melanosomes visualized by the transmission electron microscopy. Finally, we assessed the role of G-protein-coupled seven-transmembrane-domain receptors. Obtained results revealed that nonselective MT1 and MT2 receptor antagonist (luzindole) or selective MT2 receptor antagonist (4-P-PDOT) did not affect dysregulation of the melanin pathway indicating a receptor-independent mechanism. Our findings, together with the current state of the art, provide a convenient experimental model to study the complex relationship between metabolites of melatonin and the control of pigmentation serving as a future and rationale strategy for targeted therapies of melanoma-affected patients.

## 1. Introduction

In recent years, significant advances in our understanding of melanogenesis and its control have been made as the result of studies on melanoma [1]. Recently, numerous investigations described melanogenesis as a complex, multistage process involving the synthesis of melanosomes, the transport of melanosomes to the dendrite tips of the melanocytes, and their release. In melanoma cells and melanocytes, melanin synthesis is regulated by a cascade of enzymatic reactions. The initial step of melanin synthesis involves the oxidation of tyrosinase to 3,4-dihydroxyphenylalanine (DOPA) by tyrosinase (TYR), the rate-limiting enzyme of melanogenesis [2]. Melanogenesis is stimulated by various effectors, including paracrine melanogenic factors (α-melanocyte-stimulating hormone (α-MSH), endothelin-1, and stem cell factor), cyclic adenosine monophosphate (cAMP)-elevating agent (forskolin or cholera toxin), and ultraviolet radiation [3,4]. Also, melanogenesis is controlled via intracellular signaling pathways such as protein kinases (protein kinase A (PKA), protein kinase C-α (PKC-α), protein kinase C-β (PKC-β)) and mitogen-activated protein kinase (MAPK) [4,5,6,7,8,9]. The cAMP pathway, in particular, plays a key role in the regulation of melanogenesis through the upregulation of the key transcription factor microphthalmia-associated transcription factor (MITF) and subsequent melanogenic enzymes including pre-existing TYR protein and TYR mRNA [5,10,11]. In addition, melanogenesis can profoundly affect the biologic behavior of melanoma cells as well as that of the surrounding environment [12,13]. Hormonal and nutritional factors regulate these biochemical processes, and the concentration of these factors determines the amount of melanin production [13].

Melatonin (*N*-acetyl-5-methoxytryptamine, MEL) is the major biologically active molecule secreted by the pineal gland [14,15]. Melatonin and its metabolites have been found throughout the evolutionary spectrum, including animals, plants, microbes [16,17,18,19] and even honey [20,21]. In mammals, the pineal gland secrets melatonin into the blood stream, where it exerts a range of well-documented physiological functions, acting as a neurotransmitter for hormonal and immunomodulatory actions [22,23]. In the blood, the nighttime concentration of melatonin is in the low nanomolar range, but in other fluids and tissues, the concentration of this indoleamine is several orders of magnitude higher than that in the plasma [21]. Some of the physiological actions of melatonin are mediated by its interaction with two well-characterized G-protein-coupled seven-transmembrane-domain receptors MT1 and MT2, while other phenotypic effects are independent of these MT receptors [24,25]. 

Melatonin exerts a vast array of biological functions where the cell growth inhibitory properties have been one of better known actions. Thus, anti-proliferative activity has been demonstrated both in vitro and in vivo in melanoma cells [26,27,28,29,30,31]. Next to melatonin, the increasing importance of its metabolites has come into focus due to their ability to reduce melanoma growth. Indolic, kynurenic, and classical pathways are described as the main pathways of melatonin metabolism [32,33,34]. *N*^1^-acetyl-*N*^2^-formyl-5-methoxykynuramine (AFMK) is produced from melatonin via kynurenic pathway [33,34], through interaction with H_2_O_2_ [35,36]. Metabolites of melatonin also include 6-hydroxymelatonin (6(OH)MEL), and 5-methoxytryptamine (5-MT) produced via the indolic pathway [37].

We have recently reported that melatonin controls melanin synthesis in both rodents [38] and human melanoma cells [30]. In this report, we comparatively assessed the impact of its metabolites on the regulation of melanogenesis and the role of MT1 and MT2 membrane receptors in this process using the conventional receptor antagonists, luzindole and 4-P-PDOT. 

## 2. Results

### 2.1. Melatonin and Its Metabolites Decrease Tyrosinase Activity and Melanin Content 

Firstly, we assessed the impact of subjected compounds on tyrosinase activity using amelanotic (A375, Sk-Mel-28) and highly pigmented (melanotic) (MNT-1) melanoma cells (Figure 1A–F). A significant decrease in TYR was noticed in all cellular models ranging from 18% to 26% for 10^−6^ M (Figure 1A,C,E) and from 26% to 49% for a higher dose (10^−3^ M) of melatonin and its metabolites (Figure 1B,D,F). Comparatively, treatment with 10^−3^ M *N*-phenylthiourea (PTU, positive control) triggered a drop in TYR activity on average of 61% versus control cells. 

Furthermore, the evaluation of melanin content and following assessments were performed using highly pigmented melanoma cell lines. Thus, melanotic MNT-1 melanoma has been identified as an optimal model to investigate the mechanism of melanogenesis itself but also the efficacy of melanogenic regulators. This selection is also in agreement with previous reports regarding pigmentation research [39,40,41,42]. Namely, melatonin and its metabolites distinctly decreased melanin content by 15% at a dose of 10^−6^ M (Figure 2A) while a higher concentration (10^−3^ M) significantly deepened this response reaching the level versus control cells of 32% (MEL), 31% (AFMK), 23% (5-MT), and 25% (6(OH)MEL) (Figure 2B). Cells incubated with 10^−3^ M PTU revealed a collapse of melanin synthesis of 53%. 

Additionally, we evaluated the role of G-coupled membrane MT1 and MT2 receptors. Thus, their activation has been reported to be involved in melatonin’s anti-proliferative properties in human cancer cells such as colon cancer cells [43], prostate cancer cells [44,45], or melanoma cells [46,47]. Knowing the presence of MT1/MT2 melatonin’s receptors, we further examined whether melatonin-regulated melanin content and TYR activity are receptor-mediated; experiments were performed in the presence of two well-characterized melatonin receptor antagonists (luzindole and 4-P-PDOT). The obtained results indicate that neither luzindole (Figure 3A,C), nor or 4-P-PDOT (Figure 3B,D) affected the actions of melatonin or its metabolites on melanogenesis, suggesting that this process can be regulated independently of MT1 and MT2. Namely, in the presence of luzindole, there was drop of melanin content ranging from 22% (6(OH)MEL) to 37% (MEL) and a collapse of TYR activity from 37% to 51% for 6(OH)MEL and melatonin, respectively. In addition, pre-incubation with 4-P-PDOT showed similar pattern of regulation (Figure 3B,D).

### 2.2. cAMP Is Reduced by Melatonin and Its Metabolites Independently of G-Coupled Membrane Receptors

Melanin synthesis in pigmented cells is regulated by a complex and interconnected set of intracellular signaling pathways, including PKA (stimulated by cAMP elevators), PKC (stimulated by endothelin) and MAPK (ERKs; stimulated by growth factors). The cAMP plays a pivotal role in the regulation of melanogenesis through the activation of PKA and cAMP responsive element binding protein (CREB), which led to the upregulation of the expression of microphthalmia-associated transcription factor (MITF) [48]. 

First, we assessed the effect of MEL and its metabolites on the regulatory pathways of melanogenesis including cAMP signaling. MNT-1 melanoma cells revealed a significant decrease in cAMP (Figure 4A) of 45% (MEL), 53% (AFMK), 62% (5-MT) and 49% (6(OH)MEL) according to our observation of decreased TYR activity and melanin synthesis. In parallel, we assessed cAMP level in the presence of 10 μM luzindole (Figure 4B) and 0.1 μM 4-P-PDOT (Figure 4C). The obtained results showed clearly that a reduction of cAMP level is induced by MEL and its metabolites independently of the MT1/MT2 membrane receptors. Additionally, we evaluated the effect of subjected indoleamines on *MITF* expression, the cAMP downstream pathway. Thus, we observed its distinct reduction in the presence of MEL and its metabolites (Figure 4D) and the resultant loss of melanin content in melanosomes (Figure 5). 

### 2.3. In Silico Assessment

To determine whether levels of MTNR1A or MTNR1B expression might be associated with disease outcome in melanoma patients, we examined publicly available expression data from TCGA (Figure 6). The Kaplan–Meier curves indicated that there were no significant differences in overall survival between melanoma patients stratified by MTNR1A (78.97 vs. 72.06, *p* = 0.5557) and MTNR1B (74.73 vs. 79.59, *p* = 0.4250). Similarly, there we no statistically significant differences between overall survival of melanoma patients stratified by low and high expression of MTNR1A (107.40 vs. 68.09, *p* = 0.1301) and MTNR1B (93.01 vs. 72.06, *p* = 0.5898).

## 3. Discussion

Melanogenesis at the cellular level is dependent on the formation of melanosomes, which can be produced in varying sizes, numbers, and densities depending on melanin content. At the subcellular level, melanogenesis is dependent on the expression of genes and proteins such as melanogenesis-related enzymes, i.e., tyrosinase (TYR) or tyrosinase-related protein 1 and 2 (TRP1/TRP2). In order to investigate the regulation of melanogenesis, we used melatonin (MEL), its kynurenic (*N*^1^-acetyl-*N*^2^-formyl-5-methoxykynurenine, AFMK) and indolic derivatives (6-hydroxymelatonin, 6(OH)MEL and 5-methoxytryptamine, 5-MT) in human melanoma cells. In parallel to amelanotic melanoma cells (A375, Sk-Mel-28), we conducted melanin assessments using highly pigmented (melanotic) MNT-1 melanoma cells. This selection has been justified in earlier reports to examine the mechanisms underlying melanogenesis regulation and widely understood pigmentation studies [39,40,41,42]. On the other hand, melanoma is one of the fastest increasing cancers worldwide and human melanoma cell lines are valuable models to study the mechanism of melanin synthesis by physiological compounds to improve treatment. 

Melatonin displays a variety of biological properties and one of its first described effects was the lightening of skin in amphibians. It was reported that melatonin attenuates melanogenesis in cultured cells from rodent melanomas [38,49]. Other reports revealed that melatonin in human melanoma cells may both stimulate TYR activity in Sk-Mel-28 cells [47] and decrease melanogenesis in highly pigmented MNT-1 cell lines [50]. Based on this, an explanation for the differential response to melatonin treatment remains unknown and could be multifunctional. One of the reasonable explanations could be the fact that melatonin may be metabolized through indolic and kynurenic pathways targeting the appearance of AFMK, 6(OH)MEL or 5-MT in melanoma cells [51]. To date, it was well-reported that the lightening effects of melatonin on the skin of lower vertebrates and inhibition of pigmentation in some furry animals are well appreciated [4,52]. In our report, we comparatively assessed the effect of MEL, AFMK, 6(OH)MEL and 5-MT where we not only noticed significant attenuation of TYR activity and melanin content but also a reduction of cAMP and downstream MITF responses. This pattern of regulation is in accordance with earlier observations where the inhibition of melanogenesis was demonstrated in melanomas [38,53,54]. 

Kim et al. [55] showed that melatonin and some metabolites inhibited tyrosinase and the proliferation of cultured human epidermal melanocytes. Comparatively, Hardman et al. [56] have reported that the cutaneous circadian clock elements regulated melanogenesis and melanocyte activities in the human epidermis and HFs. Thus, although there are conflicting results on melatonin functions in human hair and epidermal pigmentation [4,57,58], locally produced melatonin may play a role in the regulation of melanocytic activities via its impact on the peripheral clock. Therefore, the testing of topically applied melatonin during defined circadian windows as an external modulator of intracutaneous clock activity is warranted. Also, topical supplementation of melatonin and its precursors may be beneficial by ameliorating the oxidative environment of vitiligo skin. It should be noted that in enclosed studies, we operated along subphysiological (10^−6^ M) and pharmacological (10^−3^ M) doses. This is in line with previous reports where indeed physiological concentrations (10^−8^–10^−7^ M) affected proliferation with no evident effects on melanogenesis. Contrarily, higher doses (>10^−6^–10^−3^ M) distinctly inhibited melanin synthesis which was additionally confirmed by us using electron paramagnetic resonance [50]. 

Logan and Weatherhead [59] have confirmed that melatonin leads to the inhibition of melanogenesis independently of applied series of blockers of this process. This may imply that melatonin arrests melanin synthesis via a mechanism which operates at some post-tyrosinase step in the melanin biosynthetic pathway. Slominski and Pruski [38] also reported that melatonin at higher doses acts as a competitive inhibitor rather than acting through melatonin receptors or via binding sites for ligands. Finally, MEL and its metabolites reduced the activity of TYR, which is again in agreement with earlier reports performed in human melanocytes, and this may suggest that these substances can be used as adjuvants in treatment of skin hyperpigmentation or they could attenuate malignant transformation of epidermal melanocytes [55]. Indeed, reduced melanogenesis caused by MEL and its metabolites is in line with applied melanogenesis, i.e., 10^−3^ M PTU and these results were earlier presented by Brozyna et al. [60] and later confirmed using the same inhibitor [61,62,63]. Additionally, it should be mentioned that, originally, melatonin’s function was to lighten the skin of amphibians by causing the melanin granules within the dermal melanosomes to aggregate around the nucleus of the skin cells in frogs [64]. As a result, melatonin was once used in patients affected by localized hyperpigmented skin in an attempt to reduce the pigmentation in these areas. However, this proved unsuccessful with no skin-lightening effect in humans [65]. The reason is that mammalian melanosomes, unlike those in amphibians, are more or less permanently dispersed and thus melatonin has little effect on its ability to alter pigment aggregation in the skin of mammals. Moreover, it has been suggested that melatonin can be employed effectively to inhibit progress of neoplastic disease in both animals and humans. In our study, we set out to uncover the *modus operandi* of melatonin and its metabolites targeting as potential anti-melanoma add-ons in currently applied therapies. On the other side, McElhinney et al. [57] observed no significant change in skin color among patients receiving melatonin, and no difference relative to controls. Furthermore, the authors concluded that melatonin’s effectiveness in mediating malignant melanoma growth is not related to the suppression of normal melanogenesis.

Nevertheless, our results implicate a logical pattern of regulation of melanogenesis but clearly suggest a greater complexity than we initially expected. For example, Slominski et al. [66], using in vitro and in vivo models, demonstrated that stimulation of melanogenesis leads to increased expression of HIF-1α and its subsequent translocation into the nucleus. Thus, it is possible that HIF-1α is induced by the production of intermediates of melanogenesis, including massive generation of reactive oxygen species [4,67,68]. Consequently, the initiation of melanogenesis affects the expression of multiple genes involved in regulating the behavior of melanocytes and melanoma cells, including the metabolic switch to glycolysis coordinated by HIF-1. This accompanies the changes in mitochondrial stress-related genes, immunity, angiogenesis, and cell proliferation. Again, this can be linked to our pharmacological dose (10^−3^ M) which was used earlier [69]. The authors noticed a reduction of melanin synthesis in presence of 10^−3^ M melatonin with parallel inhibition of the HIF-1α protein. This indicates the correlation between melatonin, HIF-1, and melanogenesis.

Numerous reports during the last two decades have defined melatonin and its critical role in maintaining the optimal physiology of the mitochondria [70,71,72,73,74]. The beneficial actions of melatonin at the level of the mitochondria are apparent in reference to quenching free radicals, reducing oxidative stress, limiting mitochondria-related apoptosis, maintaining the efficiency of the respiratory chain complexes, and ensuring ample ATP production [75,76]. Moreover, these regulatory actions are not unique to a single cell type but rather are applicable to every cell, plant, and animal containing mitochondria. This extraordinary ability of melatonin to preserve mitochondrial function implies that it reaches these organelles in sufficiently high concentrations to protect them from oxidizing-mediated dysfunction. It is generally accepted that melatonin exerts some of its biological effects through interaction with specific G-protein-coupled seven-transmembrane-domain receptors MT1 and MT2 which we pharmacologically blocked by applying selective antagonists, i.e., luzindole and 4-P-PDOT. We noticed that MEL as well as its kynurenic and indolic metabolites affect the cAMP-PKA-MITF pathway. This tempted us to claim that melatonin and its derivatives act not only on the MT1/MT2-dependent receptors, leading to a decrease in cAMP level and triggering the subsequent drop of MITF and arrest of melanogenesis. This was proof that the receptors for melatonin may be coupled to the suppression of the adenylate cyclase activity via an inhibitory guanosine-nucleotide-binding protein [77]. Accordingly, it should be noted that the occurrence of melatonin-binding sites in B16/F10 mouse melanoma cells has been recently described [78]. It should be added that it has been reported that the soluble adenyl cyclase (sGC)/cGMP pathway can also increase the cAMP content of cells by cross-talking between the cAMP and cGMP signaling pathways [79,80,81,82]. It was shown that cGMP probably inhibits the phosphodiesterase to reduce cAMP degradation and thereby increases the mitochondria cAMP content [83]. Nevertheless, the nature of the cAMP-mediated response uncoupled by melatonin and its metabolites remains a matter of speculation. 

## 4. Materials and Methods

### 4.1. Reagents

Melatonin (MEL), its derivatives (5-MT, 6(OH)MEL), 10,000 units penicillin and 10 mg streptomycin per mL in 0.9% NaCl, 4-phenyl-2-propionamidotetralin (4-P-PDOT), agarose, 0.1 M cacodylate buffer, ethanol, glutaraldehyde, HCl, 1 M HEPES solution (pH 7.0–7.6), *L*-DOPA, luzindole (*N*-acetyl-2-benzyltryptamine), lead citrate, Minimum Essential Medium Eagle (MEM) with low glucose (1000 mg/L), MEM non-essential amino acid solution (NEAA) (100×), NaOH, 1% OsO_4_, *N*-phenylthiourea (PTU), propylene oxide, Triton^®^ X-100, and uranyl acetate were purchased from Sigma (St. Louis, MO, USA). AFMK and 0.05% trypsin/0.53 mM EDTA solution were delivered by Cayman Chemical (Ann Arbor, MI, USA) and Thermo Fisher Scientific (Waltham, MA, USA), respectively. Fetal bovine serum, 0.05% trypsin/EDTA solution (1×), 1×PBS (pH 7.4), 200 mM *L*-glutamine solution, and AIM-V™ Medium were supplied by Thermo Fisher Scientific. 

### 4.2. Cell Culture and Treatment

Human melanoma cell lines included human melanotic MNT-1 cells acquired as a gift from Dr. Cédric Delevoye (Institute Curie, Paris, France) and human amelanotic cell models (A375, Sk-Mel-28) (American Type Culture Collection, Manassas, VA, USA). MNT-1 cells were cultured in MEM medium supplemented with 20% (*v*/*v*) heat-inactivated fetal bovine serum, 10% (*v*/*v*) AIM-V™ Medium, 2 mM *L*-glutamine solution, 10 mM HEPES solution, 1×NEAA, and 1% (*v*/*v*) streptomycin/penicillin solution. A375 and Sk-Mel-28 cells were maintained in MEM medium supplemented with 10% (*v*/*v*) heat-inactivated fetal bovine serum, 2 mM *L*-glutamine, and 1% (*v*/*v*) streptomycin-penicillin solution. Cells in the logarithmic growth phase were used in all experiments while 80–90% confluent cell monolayers were harvested with a mixture of 0.05% trypsin/EDTA solution.

Prior to treatment, melanoma cells were allowed to attach to the culture dish by being maintained in MEM culture medium for 24 h. Culture medium was replaced with fresh medium containing MEL, AFMK, 5-MT, 6(OH)MEL versus the control sample, i.e., 0.2% ethanol in culture medium. Ethanol served as a solvent, and was not toxic to the cells after 72 h. Therefore, the tested compounds were dissolved in absolute ethanol and further diluted with 1×PBS to yield 10^−2^ M stock solution. Cells were treated with final concentrations of 10^−6^ and 10^−3^ M for 72 h for the subsequent assessments described below. 

Comparatively, a nonselective MT1 and MT2 receptor antagonist or selective MT2 receptor antagonist was used, i.e., luzindole (10 μM) and 4-P-PDOT (0.1 μM), respectively, where cells were pre-incubated for 2 h before the subjected substances. 

### 4.3. DOPA Oxidase Activity of Tyrosinase

MNT-1 cells were seeded on 6-well plates (0.3 × 10^6^ cells/well) and incubated for 72 h with MEL, AFMK, 5-MT, and 6(OH)MEL followed by 2 h pre-incubation with luzindole or 4-P-PDOT. Cells were harvested, washed with 1×PBS, centrifuged at 1000× *g* for 10 min at 4 °C, and lysed with 0.5% Triton^®^ X-100 in 1×PBS on ice. The lysates were subsequently centrifuged at 16,000× *g* for 15 min at 4 °C, resultant supernatant was added to 300 μL of 5 mM *L*-DOPA in 1×PBS, and were incubated for 1 h at 37 °C. The dopachrome formation was evaluated by measuring absorbance at OD_475 nm_ using a BioTek ELx808™ microplate reader (BioTek Instruments, Inc., Winooski, VT, USA), and the results were presented as the percentage of the control sample.

### 4.4. Melanin Content Assessment 

Cells were incubated for 72 h with MEL, AFMK, 5-MT, 6(OH)MEL followed by 2 h pre-incubation with luzindole or 4-P-PDOT as described above. To determine melanin content, cells were harvested, washed with 1×PBS, centrifuged at 1000× *g* for 10 min at 4 °C, and solubilized in 1N NaOH, and incubated for 2 h at 80 °C. Samples were centrifuged at 12,000× *g* for 10 min at room temperature (RT). The absorbances were measured at OD_405 nm_ using a BioTek ELx808™ microplate reader, and results were presented as the percentage of the control sample.

### 4.5. cAMP Direct ELISA Immunoassay 

Cells (6 × 10^6^ cells), followed by 72 h culture on 100 mm dishes with subjected compounds, were washed with 1×PBS; 3 mL of 0.1M HCl was added and incubated for 20 min at RT. Cells were scrapped, samples were dissociated up and down until suspension was homogenous, and then centrifuged at 16,000× *g* for 10 min. To determine changes in the cAMP ratio, colorimetric ELISA immunoassay was used (BioVision, Inc., Milpitas, CA, USA). Briefly, 100 μL of sample was mixed with 50 μL neutralizing buffer, 5 μL acetylating mix provided by the supplier, and incubated for 10 min at RT to acetylate cAMP. Next, in order to dilute the acetylation reagents, assay buffer was added, and the resultant sample was ready to quantify cAMP. Namely, 50 μL of test sample was added to the protein G-coated 96-well plate, and subsequently the samples were incubated with 10 μL of rabbit anti-cAMP IgG and 10 μL of cAMP-HRP for 1 h at RT. Wells were washed with assay buffer, 100 μL of HRP developer and 100 μL 1 M HCl were added, and the plate was read at OD_450 nm_ using a BioTek ELx808™ microplate reader. Results were presented as the percentage of the control sample. 

### 4.6. Transmission Electron Microscopy (TEM) Assessment

Cells were seeded on 6-well plates (0.3 × 10^6^ cells/well) and incubated for 72 h with MEL, AFMK, 5-MT, 6(OH)MEL. Next, cells were harvested and centrifuged (700 r.p.m. for 5 min), washed with 1×PBS, and fixation was performed for 24 h at 4 °C using 2.5% glutaraldehyde in 0.1 M cacodylate buffer. After that, cells were washed with 0.1 M cacodylate buffer, postfixed in 1% OsO_4_ for 2 h at RT, and washed once more using distilled water. Cells were embedded in Poly/Bed^®^812 (Polysciences, Inc., Warrington, PA, USA) after dehydration in graded ethanol (50–100%) and propylene oxide. Ultrathin sections (65 nm thick) were stained with uranyl acetate/lead citrate prior to visualization using the Jeol JEM 2100 HT transmission electron microscope and assessed qualitatively in terms of melanin content. 

### 4.7. RNA Isolation, cDNA Synthesis and PCR

RNA from MNT-1 cell pellet (5 × 10^6^ cells) was extracted according to the manufacturer’s instructions using the innuPREP RNA Mini Kit (Analytik Jena, Berlin/Heidelberg, Germany). The amount of RNA was determined using BioPhotometer (Eppendorf, Hamburg, Germany). cDNA synthesis was conducted using RevertAid™ First Strand cDNA Synthesis Kit (Thermo Fisher Scientific, Waltham, MA, USA) in presence of oligo (dT) primers as follows: 65 °C for 5 min, 42 °C for 60 min, 70 °C for 5 min in Thermomixer (Eppendorf), and the resultant cDNA was stored at −20 °C prior to PCR reaction. Reactions were carried out by GoTaq^®^ PCR Master Mix (Promega GmbH, Mannheim, Germany) in the presence of primers as follows: *MITF* (Forward: *5′*-GATGTTAGAGCAGTTCCGCC, Reverse: 5′-AGGATCCATCAAGCCCAAGA; 479 bp), *GAPDH* (Forward: *5′*-AAGGTCATCCCTGAGCTGAA, Reverse: 5′-CCCCTCTTCAAGGGGTCTAC; 498 bp). Amplification was performed using 10 min initial denaturation at 95 °C followed by three-step 39-cycling of 60 s at 95 °C (denaturation), 60 s at 60 °C (annealing), and 60 s at 72 °C (extension). PCR products were separated on 1.8% agarose gel containing RedSafe™ Nucleic Acid Staining solution (iNtRON Biotechnology, Sangdaewon-Dong, Korea), and afterwards visualized using the Fusion-FX7 UV transilluminator (Vilber GmbH, Eberhardzell, Germany). Samples were standardized by amplification of the housekeeping gene glyceraldehyde phosphate dehydrogenase (GAPDH). DNA-standard O’GeneRuler™ 100 bp DNA Ladder Mix from Fermentas International (Burlington, ON, Canada) was used. 

### 4.8. In Silico Assessment

In our analyses, we also examined the prognostic significance of MTNR1A and MTRNR1B mRNA levels in The Cancer Genome Atlas (TCGA) cohort. The survival data and gene expression for the cohort of 445 skin melanoma patients were obtained from www.cBioPortal.org and UCSC Xena Browser (http://xena.ucsc.edu/; accessed on 5 June 2023). The RNA-sequencing (RNA-seq) data sets were normalized using the upper-quartile method. The data were split into low-level (with elimination of negative expression) and high-level expression groups according to cut-off points established using the Evaluate Cutpoints R package [84].

### 4.9. Statistical Analysis

Data were expressed as pooled means +standard error of the mean (S.E.M.) of at least three independent experiments. Results were normalized and expressed as a percentage of the control value. Statistically significant differences between results were determined by the univariate analysis of variance (ANOVA) or Student’s *t*-Test and appropriate post hoc analysis using GraphPad Prism 7.05 software (La Jolla, CA, USA). A *p*-value of less than 0.05 was considered statistically significant.

For in silico assessment, survival curves were plotted using the Kaplan–Meier method, and the differences were evaluated using a Gehan–Breslow–Wilcoxon test.

## 5. Conclusions

In this study, we assessed the effect of melatonin and its kynurenic and indolic derivatives on melanogenesis and the downstream pathway of this process. These results substantiate previous molecular, histochemical and biochemical studies on human cutaneous melatoninergic system [22,51,85,86,87,88,89,90]. They also provide an initial proof of concept that the presence of G-protein-coupled receptors for melatonin and its pharmacological blockage by luzindole or 4-P-PDOT does not affect the resultant decrease in melanogenesis in human melanoma cells. Thus, these findings might provide a convenient experimental model to study the complex relationship between melatonin and the control of mammalian pigmentation. Future work will be devoted to extending this study to the control of melanogenesis in normal melanocytes, whose hormonal regulation might be somewhat different to that of their malignant counterparts.

The reported data also open exciting new possibilities on the in vivo role of local melatonin synthesis and metabolism systems in the regulation of epidermal functions. These would include regulation of its barrier function, epidermal pigmentary system, and anti-carcinogenic activity. Furthermore, regulation of their endogenous production/metabolism can serve as a rational strategy for targeted therapies of melanoma patients. Thus, based on results presented here and hypothesized recently Kleszczyński and Böhm [91] and others [92,93], we are tempted to claim that melatonin, and also its metabolites, may boost commonly used BRAF/MEK inhibitors; however, these investigations still need to be carefully checked using in vitro and in vivo models.

## Figures and Tables

**Figure 1 ijms-24-14947-f001:**
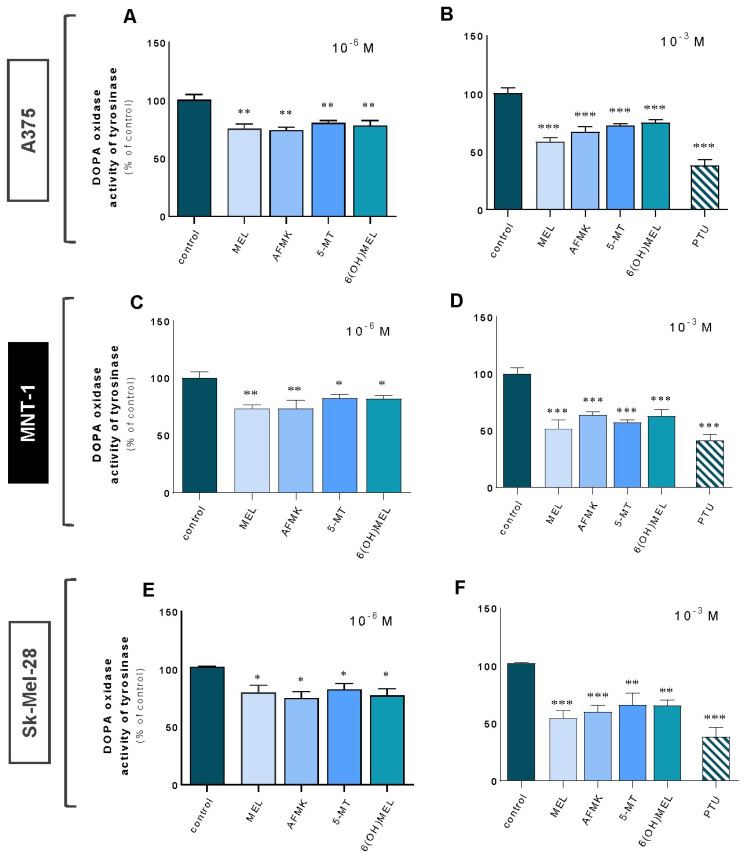
Tyrosinase activity decreased in melanoma cells in presence of melatonin and its metabolites. Amelanotic (A375 and Sk-Mel-28) and melanotic (MNT-1) human melanoma cell lines were incubated with MEL, AFMK, 5-MT and 6(OH)MEL for 72 h with 10^−6^ M (**A**,**C**,**E**) or 10^−3^ M (**B**,**D**,**F**) in comparison to melanogenic inhibitor, that is, 10^−3^ M *N*-phenylthiourea (PTU, positive control) and assessed as described in *Materials and Methods*. Data are presented as the mean + S.E.M. (*n* = 5) and values are normalized and expressed as percentage of the control value. Statistically significant differences are indicated as * *p* < 0.05, ** *p* < 0.01, *** *p* < 0.001.

**Figure 2 ijms-24-14947-f002:**
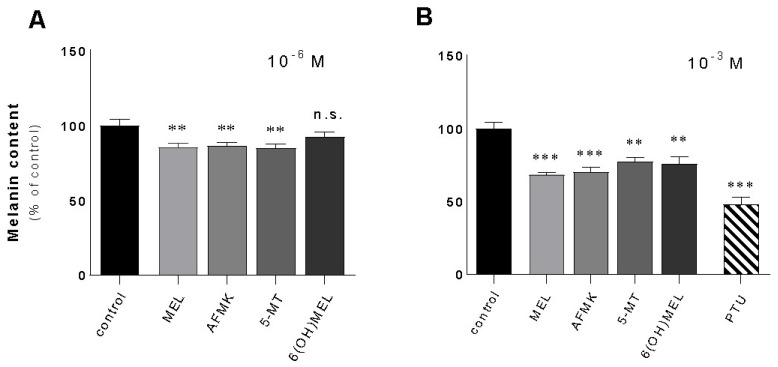
Melatonin and its metabolites reduced melanin content melanotic human MNT-1 melanoma cells. Cells were treated with MEL, AFMK, 5-MT and 6(OH)MEL for 72 h with 10^−6^ M (**A**) or 10^−3^ M (**B**) in comparison to melanogenic inhibitor, that is, 10^−3^ M PTU (positive control) and assessed as described in *Materials and Methods*. Data are presented as the mean + S.E.M. (*n* = 5) and values are normalized and expressed as percentage of the control value. Statistically significant differences are indicated as ** *p* < 0.01, *** *p* < 0.001, n.s.—not significant.

**Figure 3 ijms-24-14947-f003:**
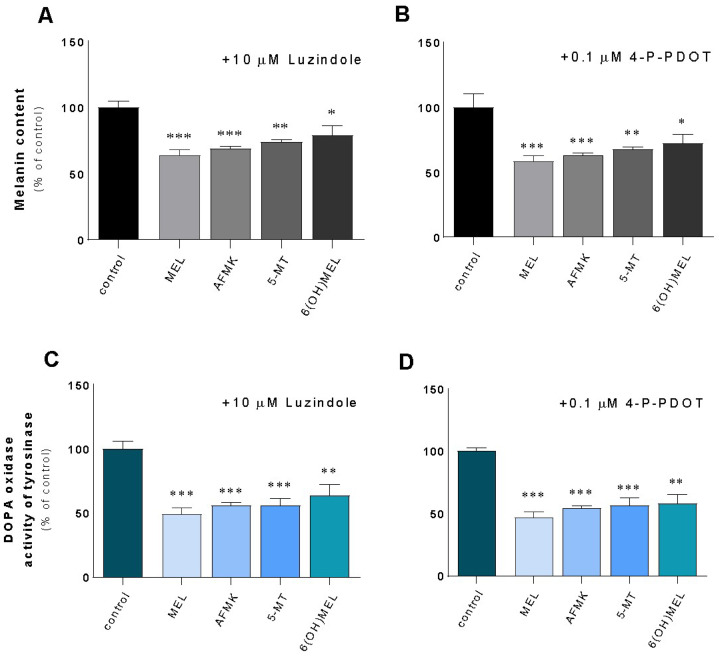
Assessment of the role of MT1/MT2 receptors in terms of melatonin and its metabolites-mediated melanogenesis in human MNT-1 melanoma cells. Cells were pre-incubated for 2 h with a nonselective MT1 and MT2 receptor antagonist, that is, 10 μM luzindole or selective MT2 receptor antagonist, i.e., 0.1 μM 4-P-PDOT. Next, cells were treated for 72 h with 10^−3^ M MEL, AFMK, 5-MT or 6(OH)MEL and assessed for melanin content (**A**,**B**) and tyrosinase activity (**C**,**D**) as described in *Materials and Methods*. Data are presented as the mean + S.E.M. (*n* = 5) and values are normalized and expressed as percentage of the control value. Statistically significant differences are indicated as * *p* < 0.05, ** *p* < 0.01, *** *p* < 0.001.

**Figure 4 ijms-24-14947-f004:**
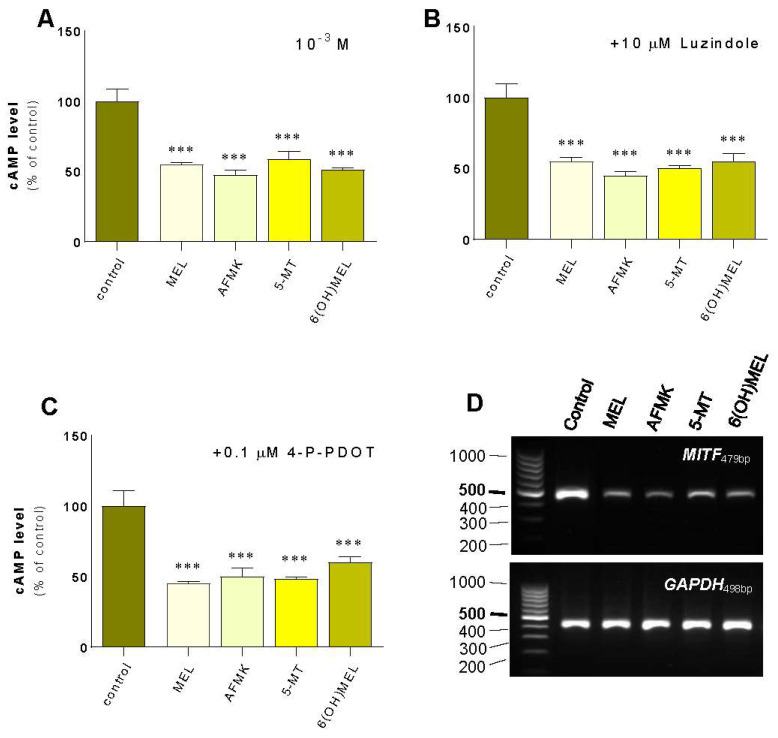
Evaluation of the effect of melatonin and its metabolites on cAMP level and *MITF* in comparison to the role of MT1/MT2 receptors. Human MNT-1 melanoma cells were treated for 72 h with 10^−3^ M MEL, AFMK, 5-MT or 6(OH)MEL and assessed for cAMP level (**A**). Comparatively, the role of MT1/MT2 receptors was assessed followed by 2 h pre-incubation with 10 μM luzindole (MT1/MT2) (**B**) or 0.1 μM 4-P-PDOT (MT2) (**C**) as described in *Materials and Methods*. Data are presented as the mean + S.E.M. (*n* = 5) and values are normalized and expressed as percentage of the control value. Statistically significant differences are indicated as *** *p* < 0.001. (**D**) PCR assessment of *MITF* expression in presence of MEL and its metabolites.

**Figure 5 ijms-24-14947-f005:**
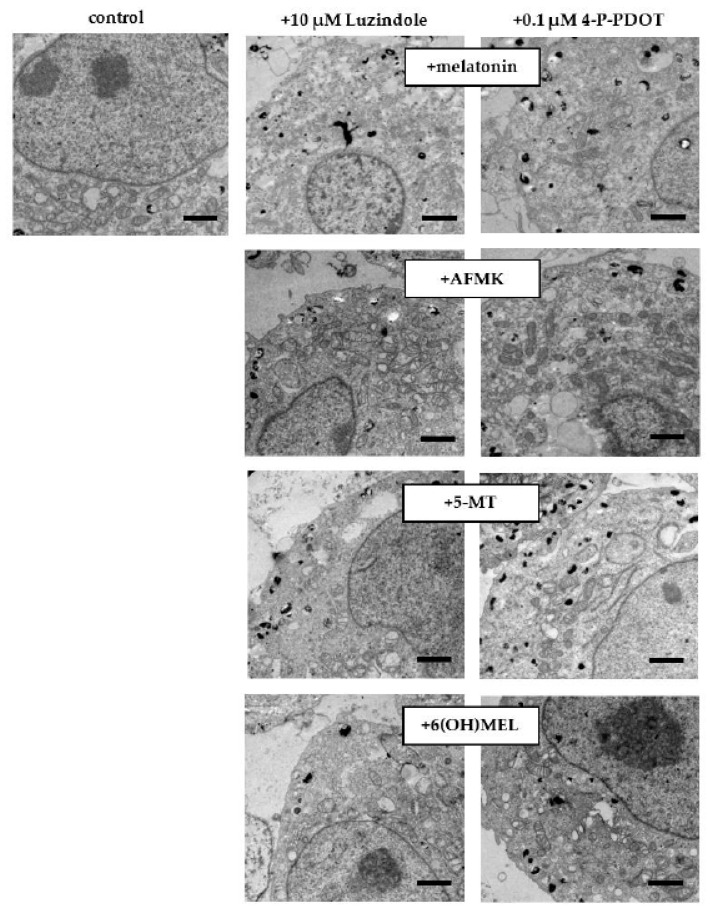
Melatonin and its metabolite-induced drop in melanin content in melanotic MNT-1 melanoma cells. After 72 h incubation with melatonin and its metabolites (10^–3^ M), transmission electron microscopy (TEM) images were obtained as described in *Materials and Methods,* where the effects of G-coupled membrane receptors (10 μM luzindole or 0.1 μM 4-P-PDOT) were assessed, and their presence did not affect the collapse of melanogenesis. Bars: 1 μm.

**Figure 6 ijms-24-14947-f006:**
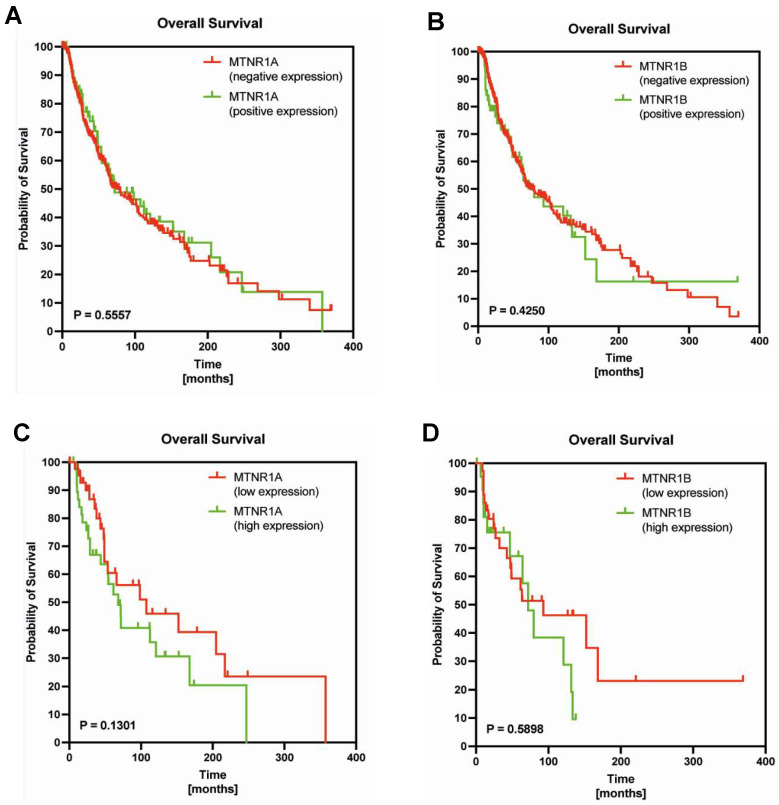
Kaplan–Meier survival curves and Gehan-Breslow-Wilcoxon test for overall survival of melanoma patients stratified by positive and negative MTNR1A mRNA expression (**A**), positive and negative MTNR1B mRNA expression (**B**), low and high MTNR1A mRNA expression (**C**), low and high MTNR1B mRNA expression (**D**) as described in *Materials and Methods*.

## Data Availability

Not applicable.

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
