# Peer review of "Melanogenesis Is Directly Affected by Metabolites of Melatonin in Human Melanoma Cells"

_ijms, 2023, doi:10.3390/ijms241914947_

Round 1
Reviewer 1 Report
1. It would be better to discuss whether the anti-melanogenesis effects of melatonin and its metabolites occur only in melanoma cells or also in normal skin cells since the literature has documented that melatonin would not lighten the skin of mammals.
2. It is well documented that melatonin is synthesized in the mitochondria and majority of its actions may also occur at the level of mitochondria (Reiter RJ, et al. Mitochondria: Central Organelles for Melatonin's Antioxidant and Anti-Aging Actions. Molecules. 2018. PMID: 29495303; Tan DX, et al. Melatonin: Both a Messenger of Darkness and a Participant in the Cellular Actions of Non-Visible Solar Radiation of Near Infrared Light. Biology (Basel). 2023. PMID: 36671781.). This novel information on melatonin research should be discussed in the text to improve the quality of the paper.
3. It is very interesting that the inhibitory effects of melatonin and its metabolites on melanogenesis are independent on its membrane receptors (MT1 and MT2) but melatonin lowers the cAMP level. It should be discussed whether this cAMP lowing activity of melatonin is mediated by directly inhibiting the adenyl cyclase or by promoting the phosphodiesterase which degrades the cAMP.
English is fine
Author Response
Reviewer #1:
- It would be better to discuss whether the anti-melanogenesis effects of melatonin and its metabolites occur only in melanoma cells or also in normal skin cells since the literature has documented that melatonin would not lighten the skin of mammals.
- It is well documented that melatonin is synthesized in the mitochondria and majority of its actions may also occur at the level of mitochondria (Reiter RJ, et al. Mitochondria: Central Organelles for Melatonin's Antioxidant and Anti-Aging Actions. Molecules. 2018. PMID: 29495303; Tan DX, et al. Melatonin: Both a Messenger of Darkness and a Participant in the Cellular Actions of Non-Visible Solar Radiation of Near Infrared Light. Biology (Basel). 2023. PMID: 36671781.). This novel information on melatonin research should be discussed in the text to improve the quality of the paper.
- It is very interesting that the inhibitory effects of melatonin and its metabolites on melanogenesis are independent on its membrane receptors (MT1 and MT2) but melatonin lowers the cAMP level. It should be discussed whether this cAMP lowing activity of melatonin is mediated by directly inhibiting the adenyl cyclase or by promoting the phosphodiesterase which degrades the cAMP.
Authors’ response: The authors express their gratitude for these valuable remarks and we additionally discussed the pointed out issues. Simultaneously, we revised the references with updated citations.
Reviewer 2 Report
The results obtained are relevant in the field of melanoma research.
The methodology applied is correct.
In Figure 3, on Evaluation of the role of MT1/MT2 receptors in terms of melatonin and its metabolites, the micromolar units are in bold and would look better if they were not in bold. Exactly the same happens in Figure 4, for images C and D.
The bibliography used is accurate, as well as necessary.
The presentation is very good and the content of the review is of relevance so I highly recommend the publication of that article.
Congratulations on the investigation!
Author Response
The results obtained are relevant in the field of melanoma research.
The methodology applied is correct.
In Figure 3, on Evaluation of the role of MT1/MT2 receptors in terms of melatonin and its metabolites, the micromolar units are in bold and would look better if they were not in bold. Exactly the same happens in Figure 4, for images C and D.
Authors’ response: We want to thank for this remark. We corrected the pointed out labelling across the all figures accordingly.
The bibliography used is accurate, as well as necessary.
The presentation is very good and the content of the review is of relevance so I highly recommend the publication of that article.
Congratulations on the investigation!
Authors’ response: The authors would like thank very much for nice comments about the report.